# DNA Gyrase as a Target for Quinolones

**DOI:** 10.3390/biomedicines11020371

**Published:** 2023-01-27

**Authors:** Angela C. Spencer, Siva S. Panda

**Affiliations:** Department of Chemistry and Physics, Augusta University, Augusta, GA 30912, USA

**Keywords:** quinolones, DNA gyrase, molecular docking, drug development, drug-resistance

## Abstract

Bacterial DNA gyrase is a type II topoisomerase that can introduce negative supercoils to DNA substrates and is a clinically-relevant target for the development of new antibacterials. DNA gyrase is one of the primary targets of quinolones, broad-spectrum antibacterial agents and are used as a first-line drug for various types of infections. However, currently used quinolones are becoming less effective due to drug resistance. Common resistance comes in the form of mutation in enzyme targets, with this type being the most clinically relevant. Additional mechanisms, conducive to quinolone resistance, are arbitrated by chromosomal mutations and/or plasmid-gene uptake that can alter quinolone cellular concentration and interaction with the target, or affect drug metabolism. Significant synthetic strategies have been employed to modify the quinolone scaffold and/or develop novel quinolones to overcome the resistance problem. This review discusses the development of quinolone antibiotics targeting DNA gyrase to overcome bacterial resistance and reduce toxicity. Moreover, structural activity relationship (SAR) data included in this review could be useful for the development of future generations of quinolone antibiotics.

## 1. Introduction

For over 100 years, antibiotics have been used to clinically treat diseases, beginning in the 1910s with salvarsan, a drug designed by Paul Ehrlich to combat syphilis [1]. However, over time, antimicrobial-resistant strains emerged and by 2019, antimicrobial-resistant pathogens were responsible for more than 4.95 million deaths, including 1.27 million deaths specifically attributable to bacterial antimicrobial resistance. This resistance is one of the leading public health threats of the 21st century [2]. A study on antimicrobial resistance by the UK government predicts that antimicrobial resistance could be responsible for killing 10 million people per year by 2050 [3]. Recently, the U.S. Center for Disease Control (CDC) estimated that over 3 million Americans acquire an antimicrobial-resistant infection each year [4]. Additionally, secondary bacterial infections are significantly more complicated when the infection is associated with COVID-19 [5], resulting in higher mortality rates for COVID-19 patients compared to non-COVID-19 patients [6]. Based on the critical need to combat antimicrobial resistance, it is no surprise that as of 2020, the market size of antibiotics was over USD 37 billion and is expected to cross USD 45 billion by 2028 [7].

Since the first use of salvarsan, the discovery of antibiotics derived from nature, fungi, or bacteria, and the development of synthetic antibacterials, has paved the way for modern medical revolution. A closer look at the mechanism of action of antibiotics derived from nature reveals common molecular targets (Table 1). Beta-lactams, glycopeptides, and other drugs, target disruption of the bacterial cell wall. Macrolides, oxazolidinones, streptogramins, and lincosamides target protein synthesis at the level of the 50S ribosomal subunit, while tetracyclines and aminoglycosides target the 30S small ribosomal subunit. Ansamycins and lipiarmycins inhibit nucleic acid synthesis at the level of RNA polymerase. While some synthetic antibacterial agents share similar targets with natural products, early drugs like sulfonamides, salicylates, sulfones, and later diaminopyrimidines, expand the list of molecular targets to include folate synthesis. Nitrofurans, azoles, and quinolones (the subject of this report) induce DNA damage, with quinolones specifically targeting the topoisomerases DNA gyrase and topoisomerase IV [1].

Topoisomerases maintain the topological state of DNA that is constantly being manipulated by cellular processes such as replication, transcription, and recombination. Eubacterial DNA gyrase is a member of the type II subfamily of DNA topoisomerases. Topoisomerases in the type II subfamily are characterized by the generation of a double-stranded break in the DNA. DNA gyrase falls into the class of type IIA topoisomerases, as does eubacterial topoisomerase IV (topo IV) and eukaryotic DNA topoisomerase II (topo II). Structurally and mechanistically, DNA gyrase and topo IV are similar, however, DNA gyrase is unique in its ability to generate negative supercoils driven by ATP hydrolysis [8], while the action of topo IV results in the ATP-driven decatenation of DNA [9]. Topo II action leads to ATP-dependent relaxation of DNA [10].

Drugs that target DNA gyrase and topo IV act to impede the catalytic activity of the topoisomerase enzymes, which, in some cases, can lead to fragmentation of the genome. When these drugs cause potentially lethal double-stranded DNA breaks, they are called topoisomerase poisons. Quinolones are DNA gyrase and topo IV inhibitors, as they impair the catalytic activity of the enzymes. In addition, because quinolone action can lead to permanent double-stranded breaks in DNA, in this mode, they are also gyrase poisons, which is the most effective among the modes of action [11,12,13,14,15]. In this review, we are focusing on quinolone-based antibacterial agents.

## 2. DNA Gyrase Structure

DNA gyrase is a heterotetramer, consisting of 2 GyrA and 2 GyrB subunits, arranged in a three-domain structure with two-fold symmetry. The B subunits of DNA gyrase contain the ATP-binding and hydrolysis sites, whereas the A subunits are responsible for the binding and wrapping of DNA [16]. Additionally, the A subunits contain the active-site tyrosine residues, located in a helix turn helix (HTH) motif within a catabolite activator protein (CAP)-like domain. Gyrase B contains a TOPRIM domain, that appears to play a key role in catalysis, by providing a binding site for the divalent cations involved in DNA cleavage and re-ligation. In generating the phosphodiester bond breakage, the 5′ end of DNA is covalently attached to the enzyme via the active site tyrosines, generating a ‘cleaved complex’ [17].

## 3. DNA Gyrase Mechanism of Action

The mechanism of action of topo II enzymes involves a staggered double-stranded break in duplex DNA. Three key structural features play important mechanistic roles: the N-gate, DNA-gate and C-gate (Figure 1). At the DNA-gate, a gated segment of DNA (commonly named the G-segment) binds, and both strands are cleaved and subsequently pulled apart by a conformational change, creating an opening [16,17]. A second DNA duplex, coming from the same or different strand (the transported, or T-segment) is trapped by the closing of the N-gate when ATP binds. The opening of the DNA-gate allows the T-segment to pass through the opening in the G-segment, moving the T-segment from an upper to a lower cavity. Re-ligation of the G-segment, followed by the release of the T-segment from the C-gate, completes the process. For the enzyme to turn over, ATP hydrolysis is required, which re-opens the N-gate, and the product is released [16,17].

In order to not only relax DNA, but also create negative supercoils, the DNA bound to gyrase is wrapped around the enzyme in a right-handed twist, a role played by the C-terminal domains (CTDs) of the A subunit. In DNA gyrase, the CTDs extend out from the body, binding and wrapping DNA in a positive manner. Based on the spatial relationship between the T and G-segments and how they are crossed, a positive node is converted into a negative node, with a decrease in the linking number by two [17]. At least one CTD is required for the introduction of negative supercoils into DNA [18,19,20]. This role of DNA gyrase is critical in reducing torsional stress, as a result of overwinding in front of replication forks and transcription complexes [17,21,22,23].

## 4. Targeting DNA Gyrase

Nalidixic acid, considered a founding member of the quinolone class of antibacterial agents, was introduced to the market as an antibiotic in the year 1962 [26]. Quinolones are broad-spectrum antibacterial agents that target DNA gyrase and topo IV [8]. Quinolones have proven to be effective in combating a variety of infections, including, but not limited to, urinary tract infections, sexually transmitted diseases, chronic bronchitis, and tuberculosis. While DNA gyrase and topo IV are the targets of quinolones in different bacterial strains, the primary targeting is based on both the bacterial species and the quinolone drug [27]. Based on an analysis of *Escherichia coli* strains carrying drug resistance mutations in both DNA gyrase and topo IV enzymes, its study concluded that gyrase is the primary toxic target for quinolones and that topoisomerase IV is a secondary drug target [28]. It appears that in most cases, the primary target of fluoroquinolones in Gram-negative bacteria is gyrase, whereas Topo IV is their preferential target in Gram-positive bacteria [29,30,31,32,33]. Figure 2 provides structural information for standard drugs **S1–S9** that act against DNA gyrase by forming multiple quinolone–enzyme–DNA complexes [34,35]. With the importance of the quinolone scaffold on its antibacterial properties, here we focus on the impact of quinolones on DNA gyrase.

Quinolones act by stabilizing the gyrase–DNA cleavage complex [36,37]. Quinolones have both bacteriostatic and bactericidal actions [23]. The stabilization of gyrase–DNA complexes stalls replication forks and slows bacterial growth. At higher concentrations, quinolones become bactericidal as chromosomes are fragmented, and cells rapidly die [37]. Quinolones act by interacting with both the DNA and the gyrase. Two drugs bind to the gyrase heterotetramer in a parallel fashion according to crystallographic data (Figure 3) [38]. Quinolones bind non-covalently to DNA gyrase in the active site and via stacking interactions with the DNA bases on either side of the site to be cleaved [39]. These binding events stabilize the gyrase–drug cleavage complex and ultimately inhibit the re-ligation of the DNA [40]. As such, quinolones act as DNA gyrase poisons, in addition to acting as catalytic inhibitors [41,42].

Quinolone binding to DNA gyrase is mediated through a water–metal ion bridge in the A subunit of DNA gyrase [40]. Specifically, the C-3/C-4 keto acid region of the quinolone is chelated by a noncatalytic Mg^2+^ ion that is also coordinated by four water molecules. The coordinated waters form hydrogen bonds with Ser83 and a nearby acidic residue, Asp87 (*E. coli* GyrA numbering) [43]. Not surprisingly, common mutations in DNA gyrase observed in drug-resistant strains are found at residues forming part of the water–metal ion ‘bridge’ between the enzyme and the drug [23,44,45]. In fact, the region between amino acids 67–106 in GyrA is referred to as the quinolone resistance-determining region (QRDR) [46]. Mutation of both the serine and the acidic residue renders the quinolone ineffective in binding and/or inhibiting the enzyme, providing evidence that these residues are crucial in forming the bridge between the quinolone and the enzyme [47]. In particular, a mutation in GyrA(Ser^83^→Trp) gives ≈ 20-fold resistance to a wide range of quinolones [48].

Based on crystallographic studies, the C-7 ring system of quinolones extends into the B subunit of DNA gyrase, where residues form a favorable, but non-specific environment for the C-7 moieties [38]. Mutations in the B subunit of gyrase have also been associated with drug resistance [37].

## 5. Synthetic 4-Quinolones Targeting DNA Gyrase

Quinolones are broad-spectrum antibiotics with a nitrogen-containing bicyclic scaffold, modified by various substitutions which play a critical role in their antibacterial properties (Figure 4). Various research and review articles have been published on quinolones and their antibacterial, as well as other biological properties [49,50]. The previous review articles were focused either on antimicrobial drugs/molecules with different targets [51,52,53,54,55], or molecules effective against a specific target, such as protein synthesis [56,57], cell wall [58,59], and DNA gyrase [60,61,62,63]. Although many have covered the breadth needed as well as the importance of quinolones in drug discovery [64,65,66,67,68], we could not locate any reports which exclusively focus on the DNA gyrase inhibitory properties of quinolone and the detailed insights. This review aims to provide in-depth details of reported quinolones targeting DNA gyrase, and their relationship with structural activity. In addition, the emergence of bacterial resistance to quinolone antibiotics (Table 1, Entry **8**) and the synthetic strategies implemented to combat this resistance, will be discussed.

### 5.1. 4-Quinolones with Free -COOH Group at C-3

Nalidixic acid is generally considered to be the first quinolone antibiotic, however, because of its narrow spectrum of activity, its use was limited to urinary tract infections [60]. Second-generation quinolones included fluorine at the C-6 position, which greatly increased the drug’s activity [60]. Ofloxacin is a second-generation fluoroquinolone used for both Gram-positive and Gram-negative bacterial infections. However, like other antibiotics, several bacterial strains developed resistance to ofloxacin. In an attempt to develop new drug candidates to combat resistant bacteria, a series of ofloxacin analogs were synthesized and screened for in vitro and in vivo antimycobacterial activities against *Mycobacterium tuberculosis* H37Rv (MTB), multi-drug-resistant *Mycobacterium tuberculosis* (MDR-TB), and *Mycobacterium smegmatis* (MC^2^). From the synthesized series, compound **1** was identified as the most potent analog with a MIC_99_ of 2.63 µM and 2.63 µM (MIC_99_ of ofloxacin is 2.16 µM and 34.59 µM) against MTB and MTR-TB, respectively. Additionally, compound **1** was found to be the most active in the inhibition of the supercoiling activity of DNA gyrase with an IC_50_ of 10.0 µg/mL. However, another compound **2** from the series shows a MIC_99_ of 0.19 µM and 0.09 µM against MTB and MTR-TB, respectively, but not inhibiting the supercoiling activity of DNA gyrase. The potency of compound **2** is probably due to another mechanism [69].



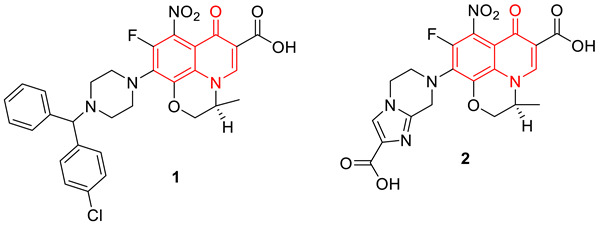



To overcome the MDR problem, new derivatives of gatifloxacin have been synthesized by Aubry et al. The newly synthesized 3ʹ-piperazinyl derivatives of the 8-hydrogeno and 8-methoxy-6-fluoro-1-cyclopropyl-4-quinolone-3-carboxylic acids were screened against pathogenic mycobacteria (*M. leprae* and *M. tuberculosis*), and wild-type strains. The MIC and DNA gyrase data conclude any variation at the 3′-position of piperazine ring reduces antibacterial properties. However, among several 3′-piperazinyl derivatives, compound **3** (with a methoxy at R8 and a secondary carbamate at R_3_′) and compound **4** (with hydrogen at R_8_ and an ethyl ester at R_3_′) showed comparable antibacterial activities as to ofloxacin [70].



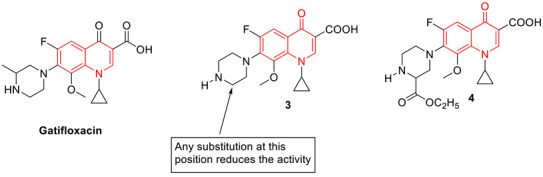



A set of benzimidazole quinolones were synthesized by Zhang et al., using a hybridization approach. Most of the synthesized hybridized compounds show higher antimicrobial potential, especially against MRSA (MIC: 0.125 µg/mL), even superior to the reference drugs (chloromycin, norfloxacin, ciprofloxacin, and ciprofloxacin). Compound **5** was found to be the most active among all the synthesized hybrids. In addition to antibacterial properties, compound **5** also inhibited the formation of biofilm and interrupted the established *Staphylococcus aureus* and *Escherichia coli* biofilms. Compound **6** showed low toxicity toward normal mammalian cells. Further, molecular docking studies suggest compound **5** binds DNA effectively and forms a stable complex that might block DNA replication and exert potent bioactivities [71].



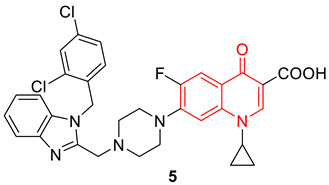



Fluoroquinolones are well known for their interaction with topoisomerase and DNA, after the two DNA strands are cleaved and remain covalently attached to the active site tyrosine. The computer-aided drug design approach was considered by Towle et al., to develop newer versions of fluoroquinolones. Their study demonstrates that regardless of potential fit in the static structure, extended N-1 groups interfere with the cleaved complex formation and poisoning. Several compounds were synthesized by extending the N1 position and investigating the binding ability to the DNA. From this approach, compounds **6** and **7** were identified as the most potent antibacterial agents with no poisoning effect, however, the detailed mechanism was unclear [72].



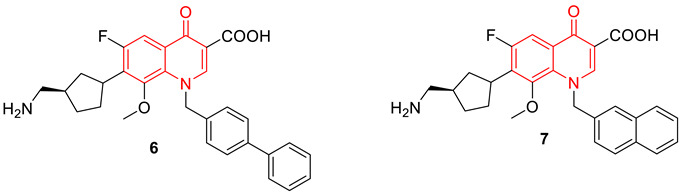



Arab et al. developed several 7-piperazinylquinolones containing a (benzo[d]imidazol-2-yl) methyl scaffold, and studied their antibacterial properties against Gram-positive and Gram-negative bacterial strains. Compound **8** proved to be the best compound of all the synthesized compounds. Compound **8** showed the highest activity against *Staphylococcus aureus*, *Staphylococcus epidermidis*, *Bacillus subtilis*, and *Escherichia coli*, with a MIC value of 0.097 μg/mL. Computational studies indicate the docking poses of compound **8** against DNA gyrase, subunits A (PDB code: 2XCT) and subunit B (PDB code: 3TTZ), are comparable with the reference standard [73].



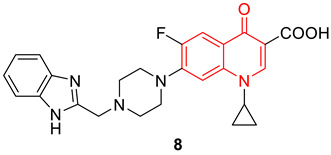



Carta et al. developed several F-triazolequinolones (FTQs) and alkoxy-triazolequinolones (ATQs) targeting *Mycobacterium tuberculosis* (*Mtb*). The screening data suggest ATQs are better antibacterial agents than FTQs. Compounds **9** and **10** were endowed with the anti-*Mtb* potency, with MIC values of 6.9 and 6.6 µM, respectively, and without showing any toxicity to the Vero cell line. Both compounds show *M. tuberculosis* DNA gyrase inhibition (IC_50_: 27–28 µM) in a DNA supercoiling activity assay. Further molecular docking studies with a 3D model structure of the *Mtb* DNA gyrase (PBD: 5BTC) confirm the interactions and formation of complex structure with compound **9** is 4.5-fold better than the reference drug ciprofloxacin. Further structure–activity relationship confirms the importance of methoxy and ethoxy groups for the potency. Surprisingly, 6-fluoro substituted analogs do not show an increase in biological activity, but rather a drastic decrease [74].



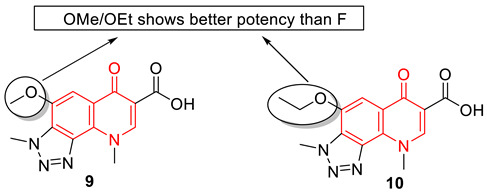



A series of fluoroquinolone-safirinium dye conjugates were synthesized from profluorophoric isoxazolones and antibiotics bearing a secondary amino group at position 7 of the quinoline ring, using well-known Mannich-electrophilic amination reactions. Synthesized conjugates were tested against several Gram-positive and Gram-negative bacterial strains, and from them, compounds **11** (conjugate of lomefloxacin) and **12** (conjugate of ciprofloxacin) were identified as the most effective ones. Having ideal lipophilicity is always a challenging task in the drug development process. Even though the synthesized zwitterionic conjugates did not show an appreciable increase in inhibition for *E. coli* DNA gyrase compared to parent drugs, they were distinctly less lipophilic than the parent quinolones in micellar electrokinetic chromatography (MECK) experiments. Evidence from molecular docking studies showed that potential conjugates could bind in the fluoroquinolone-binding mode of *S. aureus* DNA gyrase (PDB: 5CDQ) [75].



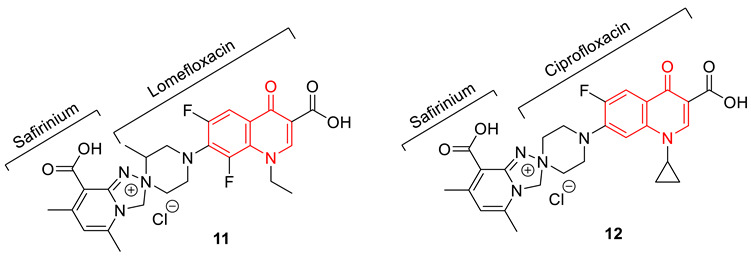



To combat bacterial resistance, a set of norfloxacin–thiazolidinedione hybrid molecules (**13**) were produced by Marc et al. The synthesized hybrids show direct activity against Gram-negative strains, and antibiofilm activity against Gram-positive strains. The antibacterial properties of the hybrids are comparable with the parent norfloxacin, however, unlike norfloxacin, have various degrees of antibiofilm activity, which were more noticeable against *S. aureus*. The MIC values against different bacterial strains are summarized in Table 2. Computational studies with DNA gyrase isolated from *Escherichia coli* (PDB: 2XCT) imply that the newly synthesized hybrids (**13**) strongly interact with both gyrase subunits (A and B) in comparison to norfloxacin [76].



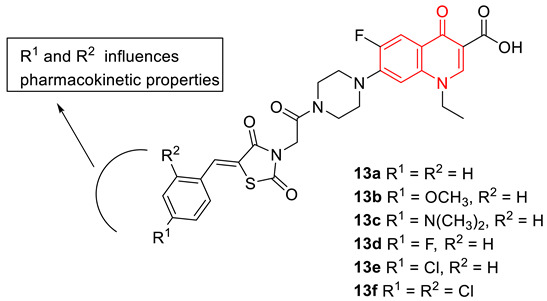



Panda et al. synthesized several fluoroquinolone conjugates from fluoroquinolone antibiotics, dichloroacetic acid (DCA), and amino acids, using a molecular hybridization approach. Among all the synthesized conjugates, compounds **14** and **15** reveal antimicrobial properties against *E. coli*, *S. aureus*, and *Enterococcus faecalis*, with potency of 1.9, 61.9, 20.7, and 2.4, 37.1, 8.3-folds, respectively, compared to the parent antibiotic (ciprofloxacin). The *E. coli* DNA gyrase supercoiling bioassay data of compounds **14** and **15** (IC_50_: 3.25 and 9.80 µM) supports the potency and possible mode of action. Chirality plays an important role in biological activity, as both compounds **14** and **15** are structurally similar, however only different at one chiral center. One is a racemic mixture (**14**) and the other is an L-isomer (**16**) [77].



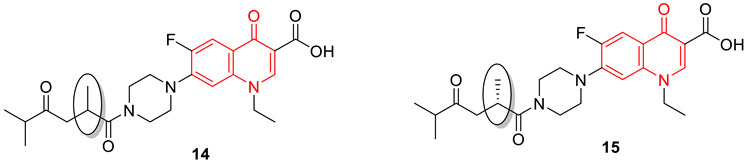



The approach of fusing two biologically active scaffolds into one heteromeric chemotype was adopted by Fan et al., to synthesize azithromycin with ciprofloxacin (**16**), and azithromycin with gatifloxacin (**17**) hybrid conjugates. Both compounds **16** and **17** show modest antibacterial properties in comparison to azithromycin, however, they show significant activity against ciprofloxacin-resistant *Staphylococcus aureus*, with MIC values of 0.076 µM and 0.14 µM, respectively. The DNA supercoiling assay and the DNA cleavage assay revealed that both compounds can poison *E. coli* DNA gyrase, although their IC_50_ values were higher than that of ciprofloxacin. The detailed experimental and computational investigation concludes the mode of action of these hybrid conjugates is a combination of the poisoning of DNA gyrase and an inhibition of protein synthesis [78].



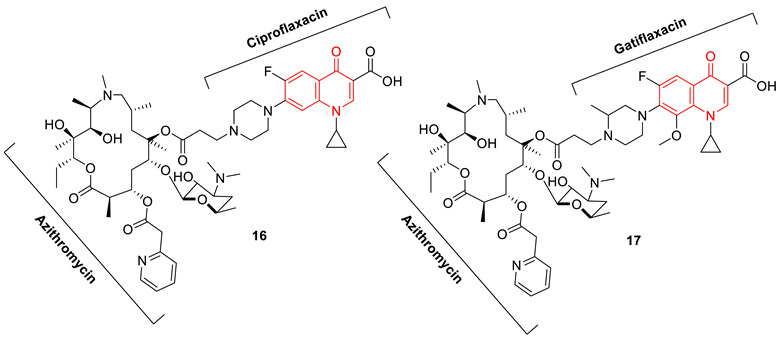



Fluoroquinolones (FQs) are considered first-line drugs for urinary tract infections (UTIs) and have been used worldwide for several decades; however, at present, FQ resistance (FQR) is a big challenge in drug development efforts. Several studies are on-going throughout the world to overcome this problem, and recently Balasubramaniyan et al. extensively utilized a 3D-QSAR approach to identify and develop potential FQ analogs (**18–27**) which showed significant antibacterial activity against FQ-resistant bacterial strains, especially FQR *E. coli*, as well as inhibitory properties against purified mutant DNA gyrase (Table 3) [79].



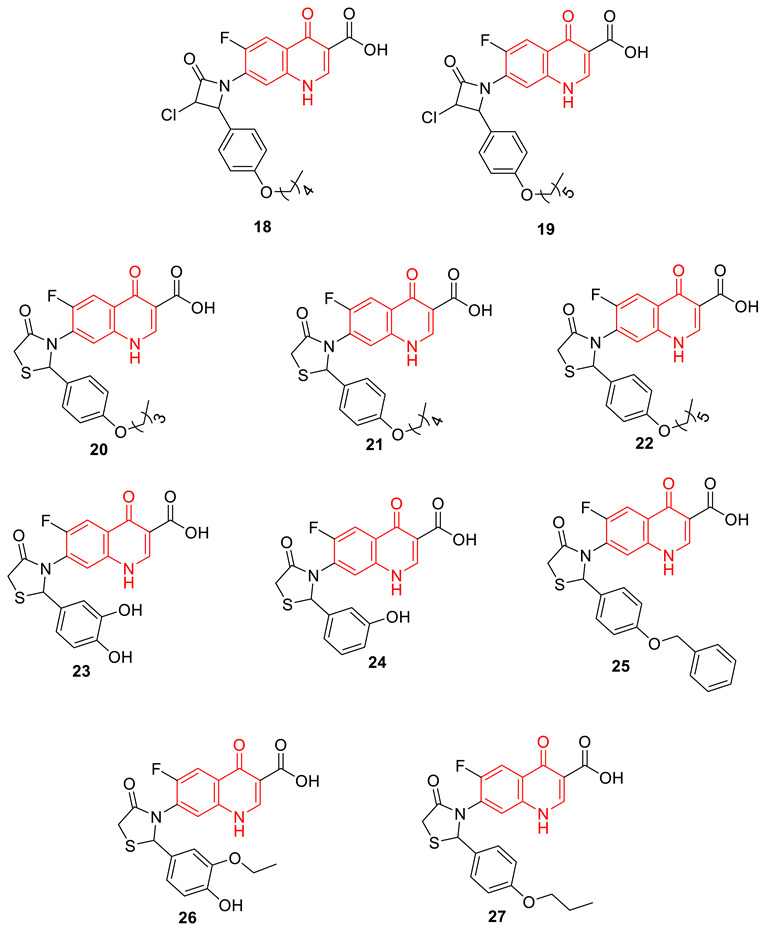



To overcome the resistance of quinolones, Chem et al. designed a new class of quinolones named 7-thiazoxime quinolones; these novel molecules contained the DNA gyrase target quinolone modified with a moiety known to disrupt the bacterial cell wall. Several analogs were synthesized and from them, compound **28** was found to be the most effective (32-fold) antibacterial agent for MRSA in comparison to ciprofloxacin. The combined use of 7-thiazoxime quinolone 28 and ciprofloxacin alleviates bacterial resistance. Mechanistic experimental investigation and molecular docking studies with DNA gyrase B (PDB ID: 3U2K) confirm that compound **28** has the ability to insert into MRSA DNA to bind with DNA gyrase, then decrease the expression of *gyrB* and *femB* genes. In addition, compound **28** is safer for mammalian cells [80].



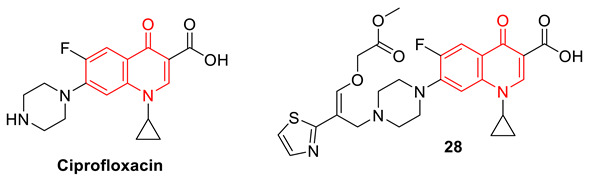



A set of water-soluble quinolones were prepared by conjugating fluoroquinolones with 4,7,10-tetraazacyclododecane-1,4,7,10-tetraacetic acid (DOTA). Among the synthesized water-soluble fluoroquinolones, compound **29** exhibited potent antimicrobial activities against MRSA and *P. aeruginosa*, with IC_50_ values of 1.56 µg/mL and 3.1 µg/mL, respectively. Atomic force microscope (AFM)-imaging investigation confirms that compound **29** could effectively destroy the MRSA bacterial membrane and cell wall. A cytotoxicity assay proved compound **29** had low toxicity to L-02, A549, and MCF-7 even at 100 µmol/L. DNA gyrase binding affinity of compound **29** was demonstrated by molecular docking studies (PDB ID: 2XCT) [81].



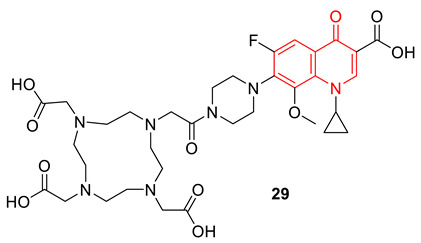



### 5.2. 4-Quinolones Lacking Free-COOH Group at C-3

Pucci et al. reported an isothiazoloquinolone (**30**) as a potential lead compound that shows significantly lower MIC_50_s and MIC_90_s against a panel of Gram-positive and Gram-negative bacterial strains, including methicillin-resistant *Staphylococcus aureus* (MRSA), using both in vitro and in vivo assays compared to standard references. The exceptional broad antibacterial property of the potential isothiazoloquinolone (**30**) is due to the dual inhibition of DNA gyrase and topoisomerase IV at low concentrations (0.68 µM and 0.12 µM) from wild-type resistant strains. Compound **3** also proved to be effective against animal infection models. In fact, the compound was effective in treating animal bacterial infections, as efficacy was observed with murine sepsis, lung, and thigh infection models [82].



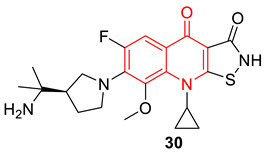



Jayagobi et al. synthesized several pyrroloquinolinone and pyrroloquinoline derivatives utilizing intramolecular domino-Knoevenagel-hetero-Diels–Alder and intramolecular imino-Diels–Alder reactions. Among all the synthesized compounds, compounds **31** and **32** show potential antibacterial properties against various Gram-positive and Gram-negative strains, with MIC values of 5 mM concentrations. These compounds also show strong DNA gyrase inhibitory properties. Compounds **33** and **34,** which are isomers of compounds **31** and **32,** are less active due to the 2-quinolone structure instead of 4-quinolone [83].



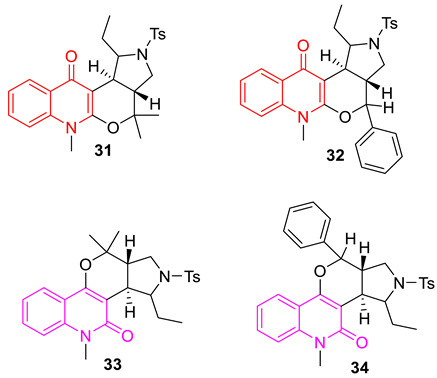



Bradbury et al. developed a series of 7-(3′-substituted) pyrrolidino-8-methoxyisothiazoloquinolone (ITQ) analogs and investigated their antibacterial properties against methicillin-sensitive *Staphylococcus aureus* (MSSA), methicillin-resistant *Staphylococcus aureus* (MRSA), and *Escherichia coli*. The antibacterial data suggest that stereochemistry plays an important role in selectivity, 7-(3′-aminomethylpyrrolidino) ITQs were generally more potent than 7-(3′-aminopyrrolidine) analogs, and that the R-isomer of the 3′-methylaminopyrrolidines was more potent (up to 16-fold) than the corresponding S-isomer. 3′-R and 1″-S configurations show higher antibacterial properties than other possible configurations. The illustration of the structural–activity relationship is depicted in compound **35**. Among all the synthesized compounds, the 7-[(R)-3-((S)-1-aminoethyl) pyrrolidin-1-yl] analog (**35**) (with MIC 0.002 and 0.06 µg/mL against MSSA and MRSA, respectively) and the (R)-7-[3-(2-aminopropan-2-yl)pyrrolidin-1-yl] analog (**36**) (with MIC 0.004 and 0.06 µg/mL against MSSA and MRSA, respectively) were found to be the best ones. The synthesized potent compounds are >30 times more inhibitory against topoisomerase IV and DNA gyrase, from both wild-type (WT) and multidrug-resistant (MDR) strains, than the fluoroquinolone moxifloxacin [84].



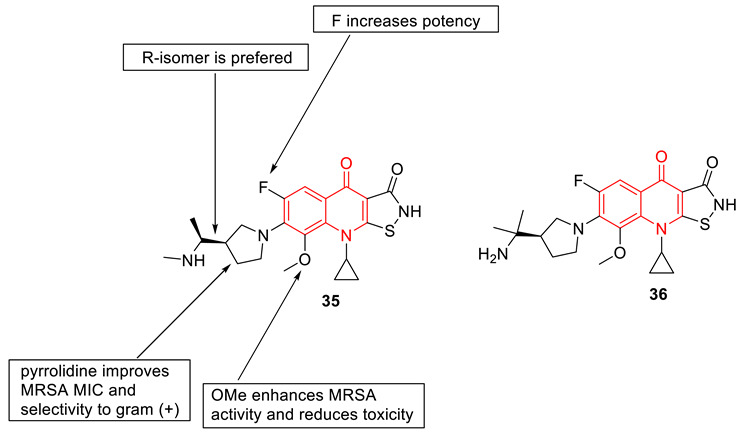



Zhou et al. designed and synthesized a set of hybrid molecules using a quinolone scaffold and 2-aminothiazole from clinical antibacterial cephalosporins to circumvent the quinolone-resistance challenge. From the synthesized various 3-aminothiazolquinolones, 3-(2-aminothiazol-4-yl)-7-chloro-6-(pyrrolidin-1-yl) quinolone (**37**) showed potential antibacterial activity against a broad antimicrobial spectrum, including multidrug-resistant strains. Compound **37** shows low toxicity to hepatocyte cells and strong inhibitory potency to DNA gyrase with an IC_50_ value of 11.5 µM in comparison to norfloxacin (IC_50_: 18.2 µM). Structure−activity relationship (SAR) studies reveal the 2-aminothiazole fragment at the 3-position of quinolone plays a crucial role in enhancing antibacterial activity. Further, molecular modeling and experimental data with DNA from a sensitive MRSA strain explain the possible antibacterial mechanism that might be associated with the formation of a ternary complex from the compound **37**−Cu^2+^−DNA, in which the Cu^2+^ ion acts as a bridge between the backbone of 3-aminothiazolquinolone and the phosphate group of the nucleic acid [85].



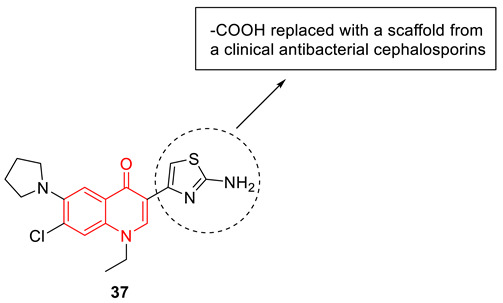



Azad and Narula developed an efficient synthetic protocol to produce 3-tetrazolyl bioisosteres (**38**) from 3-nitro derivatives of 4-quinolones, using Cu nanoparticles. Most of the synthesized 3-tetrazolyl bioisosteres showed potential antibacterial activity against several pathogenic bacterial strains, including MRSA, ranging from 12.5 to 25 µM, in comparison to ciprofloxacin (MIC: 100 µM). The observed experimental data was validated by molecular docking studies with the co-crystallized structure of the protein (DNA gyrase) in a complex with ciprofloxacin (PDB ID: 2XCT) [86].



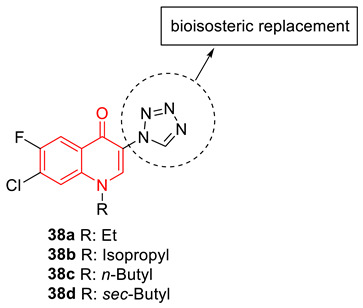



Pharmacokinetic properties are key in the drug development process. Ahmed and Kelly synthesized a set of peptide−nalidixic acid conjugates using solid-phase peptide synthesis, and investigated the role of hydrophobicity and molecular charge in improving biological activity. The peptide conjugate (**39**), with optimized hydrophobicity and molecular charges, showed substantially superior antibacterial activity. The conjugate containing cyclohexylalanine and arginine demonstrated efficient bacterial uptake and specific inhibition of *S. aureus* DNA gyrase. An organized investigation of peptide and nalidixic conjugates suggests a balance of cationic charge and hydrophobicity can overcome the intrinsic resistance of *S. aureus* DNA gyrase to quinolone-based drugs [87].



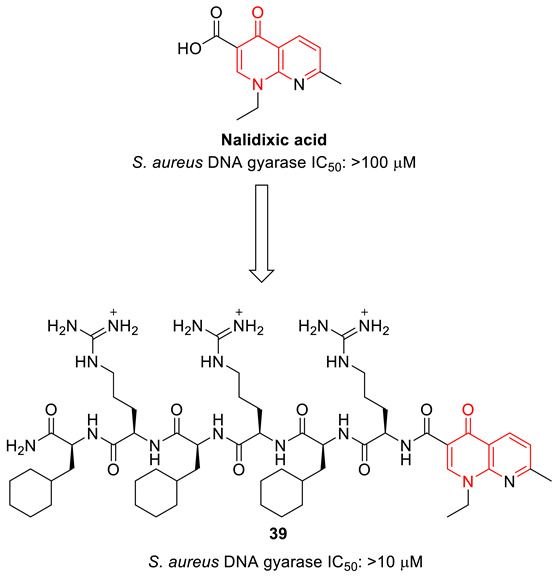



Xu et al. prepared twenty 2-sulfoether-4-quinolones using a free radical process. Most of the synthesized compounds show selective antibacterial properties against Gram-positive bacterial strains. Among all, compound **40** shows the lowest MICs against both *S. aureus* and *B. cereus* (0.8 µM and 1.61 µM, respectively). Additionally, it showed a potential inhibitory property with IC_50_ value of 0.71 µg/mL against *S. aureus* DNA gyrase. In addition, molecular docking against the gyrase–DNA–ciprofloxacin complex structure (PDB code: 2XCT) justifies the experimental data, as compound **40** docked well in the complex via precise interactions, including conventional hydrogen bonds, halogen bonds, and hydrophobic interactions. SAR suggested the introduction of a CF_3_ group enhances the antibacterial activity [88].



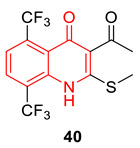



A set of multitargeting molecular hybridized aminothiazolquinolone oximes were developed from quinolone, aminothiazole, piperazine, and oxime fragments. From the synthesized hybrids, compound **41**, a C-7 substituted O-methyl oxime derivative, showed significant inhibitory efficacy against MRSA and *S. aureus* with MIC values of 0.009 mM and 0.017 mM, respectively. Toxicity studies against BEAS-2B and A549 cell lines indicate that compound **41** is safer and less likely to trigger the development of bacterial resistance. Quantum chemical studies validate the experimental data and rationally explain the structural features essential for activity. Further docking with DNA gyrase (PDB code: 4DUH) and molecular electrostatic potential (MEP) surface-studies explained the importance and interaction of O-methyl oxime fragment, thiazole ring, and quinolone scaffold. Also, drug combination studies of compound **41** with clinical antibacterial cefixime were investigated, and the observed results reveal that combined drugs were more susceptible than their individual use and their combined effects mainly exhibited synergistic and additive effects with a low MIC value of 8.72 µM (enhanced by 4-fold) against *S. aureus* [89].



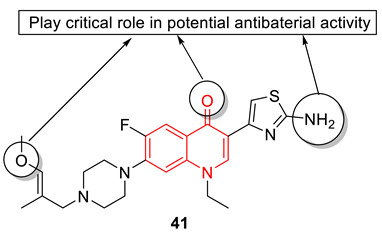



## 6. Synthetic 2-Quinolones Targeting DNA Gyrase

4-Quinolone is a well-established scaffold for antibiotics and plays a crucial role in its antibacterial properties. However, many 2-quinolones which are isomeric to 4-quinolones and isosteric to coumarins, have been investigated for various pharmacological properties, including antibacterial. The assumption is that 2-quinolone works in a similar mode of action as 4-quinolones, but more mechanistic studies are needed to confirm the antibacterial mechanism of action.

The development of non-fluoroquinolone inhibitors (4-quinolones) for bacterial infections is another area of research interest. Reck et al. systematically optimized and developed compound **42** as a potential antibacterial agent targeting quinolone-resistant isolates. One of the main goals of the development process was to enhance the IC_50_ value of the hERG. Upon introduction of F on the piperidine ring and chirality to the molecule, this increased the hERG IC_50_ value to 233 µM. The mode of action of the molecule is unclear, however, the molecule shows effectiveness in a MRSA strain-infected mouse model [90].



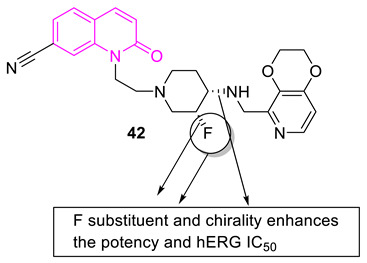



Shiroya and Patel synthesized several 2-quinolone analogs and among them, compound **43** was the most active against *S. aureus* and *E. coli*, but not as effective as ciprofloxacin. The computational study showed binding interactions within the active site of the DNA gyrase B subunit (PDB ID: 3G75) [91].



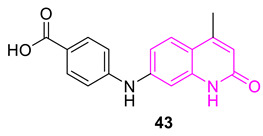



Several 2-quinolone thiosemicarbazone derivatives were prepared to start from quinolone carbaldehyde. Compounds **44**, **45**, and **46** showed moderate antibacterial properties with minimum bactericidal concentrations (MBCs) in the range of 0.80 and 36.49 mM against a broad range of bacterial strains, including MRSA. Molecular docking studies suggest these compounds are showing interactions at the active site of the DNA gyrase (PDB ID: 2XCT) [92].



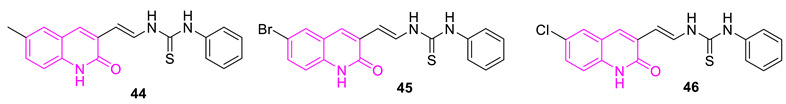



Wu et al. synthesized 37 N-thiadiazole-4-hydroxy-2-quinolone-3-carboxamides, and investigated the bacterial activity against *S. aureus*. From the study, compound **47** stands alone in terms of potency against several bacterial strains, including MRSA, by a 1 to 128-fold improvement, compared with vancomycin. It also showed low toxicity. In addition, the compound did not induce resistance development of MRSA over 20 passages, and it has been validated as a bactericidal, metabolically-stable, orally-active antibacterial agent. Further experimental (IC_50_: 0.15 µM) molecular docking (PDB ID: 4URO) data propose *S. aureus* DNA gyrase B as its potential target [93].



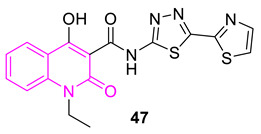



Saleh et al. synthesized several 6-hydroxyquinolinone derivatives, intending to develop potential broad-spectrum antibacterial agents. Interestingly, the intermediate **48** showed potential antibacterial properties against a broader range of bacterial strains than the final products and molecular docking studies indicate, binding to key amino acid residues of microbial DNA gyrase B of *Staphylococcus aureus* (PDB ID: 4URO) [94].



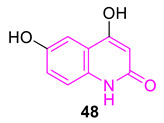



Moussaoui et al. synthesized several 2-quinolone-based compounds, incorporating triazole moiety via click chemistry. Even though some of the molecules (**49** and **50**) showed potential antimicrobial activity against *Escherichia coli*, *Staphylococcus aureus*, *Pseudomonas aeruginosa*, and *Bacillus subtilis*, molecular docking studies (PDB ID: 5BS3) suggest that the function of the synthesized 2-quinolone-based compounds was not by inhibiting DNA gyrase [95].



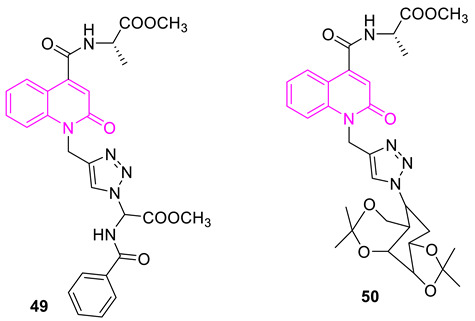



## 7. Drug-like Properties

Like the potency of a molecule, drug-like properties such as solubility, metabolic stability, toxicity, bioavailability, etc., are also equally important in the drug development process. If a molecule is potent and possesses drug-like properties (drug-likeness), then the molecule could further be considered as a drug candidate. We used a computational software “STARDROP” to determine the properties, such as molecular weight (MW), lipophilicity (logP), hydrogen bond donor (HBD), hydrogen bond acceptor (HBA), topological polar surface area (TPSA), rotatable bonds, hERG inhibition potential (hERG pIC_50_), blood–brain–barrier ability (BBB), and human intestine absorption (HIA) of all molecules discussed in this article [96].

A large molecular size/weight tends to decrease absorption and lead to a lower volume of distribution. The ideal oral drug candidates should have a molecular weight less than 500. Lipophilicity (logP) plays a crucial role in drug development, since this value is directly related to absorption, distribution, drug-binding ability, bioavailability, and drug clearance. The optimal value of logP is <5. Hydrogen-bond donors (HBD) and hydrogen-bond acceptors (HBA) are important for the interaction of drugs with the active site of the receptor. These interactions are critical for biological properties and toxicity. As per the rule of five, the number of HBD should be <5 and HBA <10. The topological polar surface area (TPSA) is a popular property of the drug development process, as this will determine the permeability of the drug. The values are different depending on the target. For non-CNS drugs, the ideal TPSA value is less than 140 Å. The number of rotatable bonds in a molecule determines its flexibility, as well as its selectivity. Generally, less than 10 rotatable bonds are most acceptable. The hERG pIC_50_ values are vital to consider, since these values indicate possible cardiac toxicity (especially compounds with >5 hERG pIC_50_ value). High hERG toxicity values cannot be tolerated in antibacterial agents. Human intestine absorption (HIA) and blood–brain–barrier ability (BBB) are considered to avoid unwanted toxicity and improve bioavailability. If the drug is not designed for the CNS, the drug should ideally not cross the BBB. The human intestine is large and a good site for absorption of most oral drugs.

The violation of drug-likeness, or Lipinski’s rule of five, including oral bioavailability, could be overcome by lead optimization/bioisosteric approaches to design and develop potential drug candidates within the applicability domain of potency and all pharmacokinetic properties. The predicted properties are listed in Table 4, and we believe these data will enable researchers to wisely choose the scaffold, and possible substituents considering drug-like properties in developing potential drug candidates to overcome the current challenges.

When developing potential oral antibacterial drug candidates, drug-like properties, including Lipinski’s rule of five, should be one of the key criteria in addition to the rationale of the drug design approach and synthesis. We have generated data on common drug-like properties of the quinolones discussed in this article. Most of the potential compounds have the required/desired parameter. However, some compounds (highlighted in red) do have antibacterial properties, but violate the recommended values of critical drug-like properties.

## 8. Methodology

The references considered for this review article were retrieved from PubMed, SciFinder, Springer, ScienceDirect, ACS, Google Scholar, and Wiley databases within the last two decades, and the search keywords used “DNA gyrase” combined with “quinolones” and further filtered by synthesis. Both experimental and computational studies for DNA gyrase investigations were reported in this review. We also used the terms “fluoroquinolone”, “gyrase inhibitor”, “quinolone-resistant”, etc., to identify missing relevant articles for inclusion in the review. The search strategy identified 582 publications, and patents were excluded from the search. We have also searched current clinical trials on quinolones as a potential therapy for bacterial infections using www.clinicaltrials.gov (accessed on 30 December 2022) [97]. Currently, there are no new molecules under the category of quinolones in the process of clinical trials.

## 9. Conclusions

Quinolones are one of the most important classes of antibiotics, however, in recent years the clinical use of these drugs is being impacted due to the growing number of resistant bacterial strains. With drug resistance emerging as a major public health concern, much drug development effort has been centered around the challenge of developing new effective drug candidates. Modifications of the quinolone scaffold have proved to overcome the resistance and enhance the potency of the drug against resistant bacterial strains. The quinolone scaffold can be modified at the N1, C3, C6, C7, and C8 (and less commonly, C5) positions to improve activity and pharmacokinetics, and reduce toxicity. Based on an analysis of the data presented in this study, the best possible substituents at each position of the quinolone are a cyclopropyl/ethyl group at N1, a fluorine/methoxy at C6, a piperazine/pyrrolidine/alkylpyrroline group at C7, and a methoxy group at C8. Fluorine at C6 was found in most of the quinolone-based compounds, as it significantly improves the activity; however, current research suggests the fluorine atom is responsible for genotoxicity.

We believe that improvements in the activity and development of potential drug candidates for resistant bacterial strains are still possible, and new generations of quinolones can still contribute to the effective treatment of bacterial infections. We hope this compiled information on quinolones can be used in the development of a new generation of quinolones with higher potency against resistant bacterial strains.

## Figures and Tables

**Figure 1 biomedicines-11-00371-f001:**
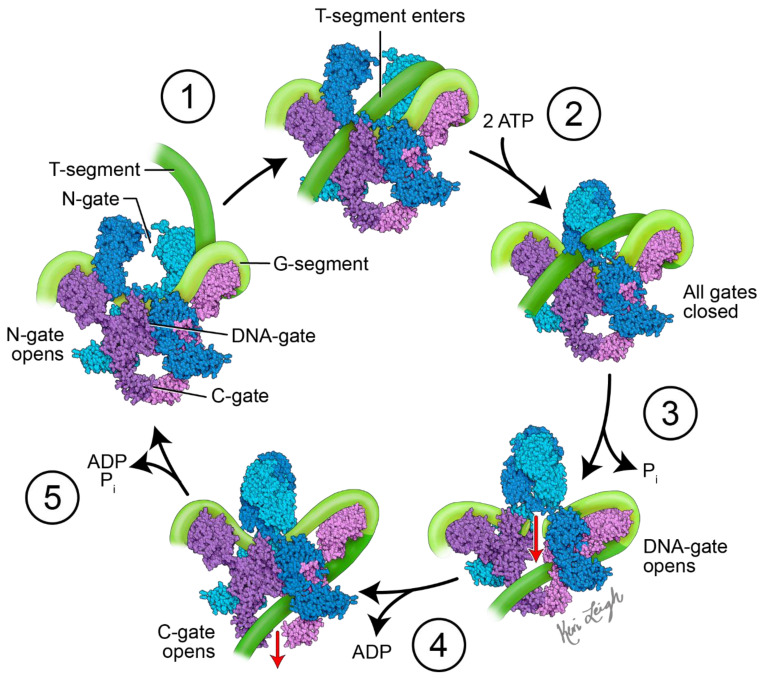
Proposed mechanism of action of DNA gyrase. Initially, DNA gyrase is bound to a G-segment of DNA at the DNA gate, a region located at the interface between subunits A (purple and pink) and B (blue and teal). The G-segment of DNA (light green) is wrapped around the C-terminal domains (CTDs) of the A subunits. The mechanistic steps are as follows: (1) the T-segment of DNA (dark green) enters the gyrase via the N-gate formed by the two B subunits. (2) When two ATP molecules bind the B subunits, the N-gate closes, and the T-segment DNA is trapped in the upper cavity of the gyrase. (3) The G-segment is cleaved, opening the DNA gate and allowing the T-segment to pass through into the lower cavity. This step requires the hydrolysis of one ATP with the release of P_i._ (4) The C-gate opens, and the T-segment passes through with the release of ADP. The second ATP is hydrolyzed. (5) The release of ADP and P_i_, along with the closing of the C-gate and opening of the N-gate, readies DNA gyrase for another cycle of supercoiling. The diagram is based on Soczek et al. [24]. The structure of DNA gyrase was derived from PDB ID 6RKW [25]. Figure created by medical illustrator Keri Leigh Jones, MSMI, CMI.

**Figure 2 biomedicines-11-00371-f002:**
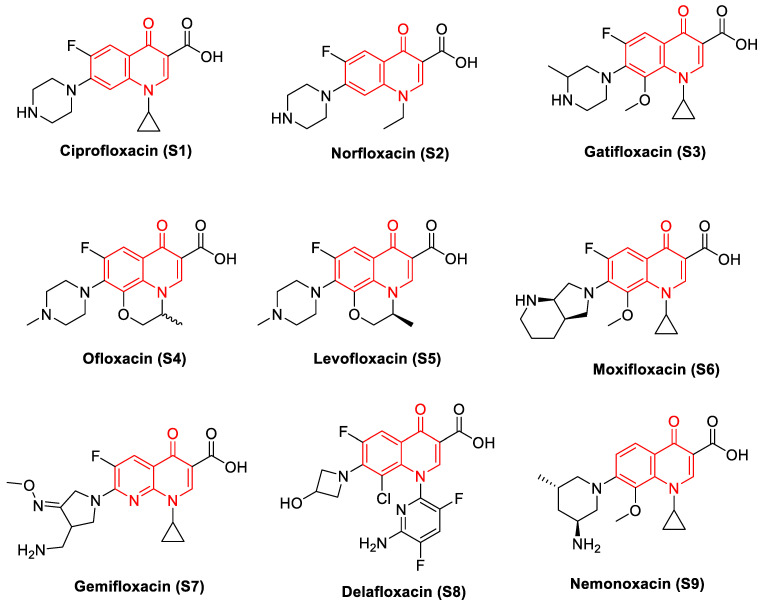
Quinolones (the scaffold colored in red): DNA gyrase poisons used as antibacterial agents.

**Figure 3 biomedicines-11-00371-f003:**
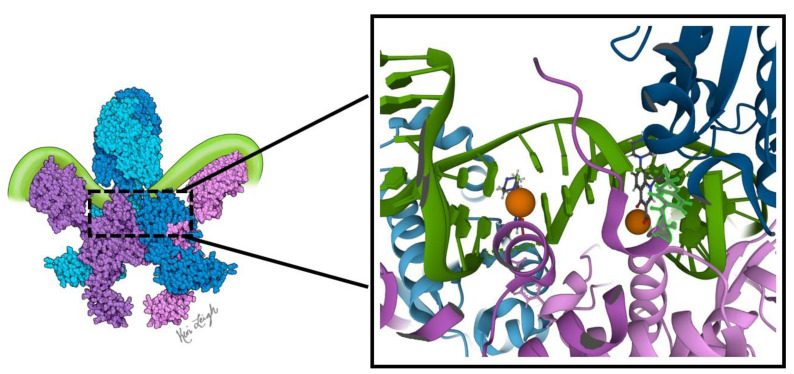
Binding of quinolones to DNA gyrase. The quinolone ciprofloxacin is shown bound to *Mycobaterium tuberculosis* DNA gyrase at the cleavage core in the dimer interface. Two quinolones (shown as ball and stick, color cpk) intercalate into the G-segment DNA (green) near the cleavage site. The noncatalytic Mg^2+^ ion that is coordinated by the keto acid group is shown in orange. The key active site tyrosines in the A subunits are shown as ball and stick in pink and purple. The structure shown on the right is adapted from PDB ID 5BTC. The color scheme is the same as in Figure 1.

**Figure 4 biomedicines-11-00371-f004:**
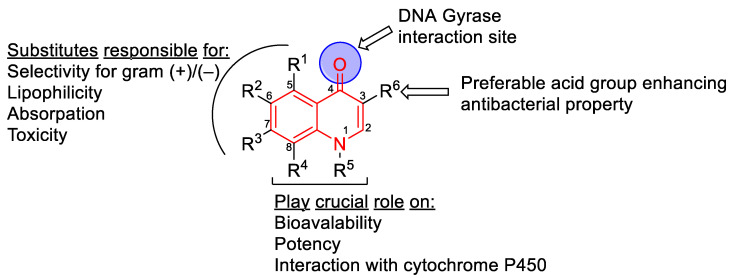
Quinolone core structure with six possible substitution locations for modification. The roles of substituents in biological activity are indicated.

**Table 1 biomedicines-11-00371-t001:** Classes of antibiotics with their mode of action.

Entry	Class	Structure/Scaffold *	Examples	Gram Coverage	Target
1	Aminoglycosides	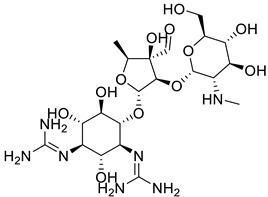	Streptomycin * Neomycin Kanamycin Paromycin Gentamicin	(−)	Protein synthesis
2	Ansamycin	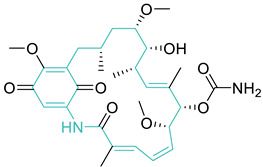	Geldanamycin * Rifamycin Napthomycin	(+)/(−)	RNA synthesis
3	*β*-Lactams	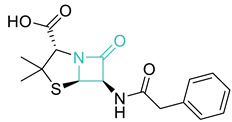	Penicillins * Penicillin G * Amoxicillin Flucloxacillin etc.	(+)/(−)	Cell wall
		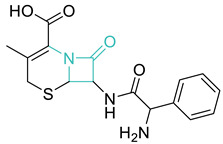	Cephalosporins Cephalexin * Cefotaxime Cefpirome etc.		
4	Glycopeptides	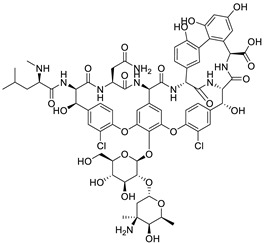	Vancomycin * Teicoplanin	(+)	Cell wall
5	Lincosamides	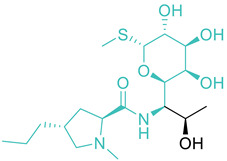	Lincomycin * Clindamycin	(+)	Protein synthesis
6	Macrolides	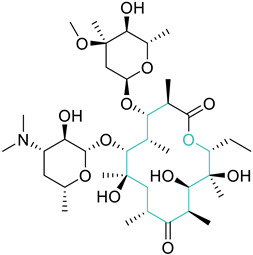	Erythromycin * Azithromycin Clarithromycin	(+)	Protein synthesis
7	Oxazolidinones	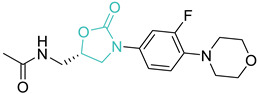	Linezolid * Posizolid Tedizolid Cycloserine	(+)	Protein synthesis
8	Quinolones	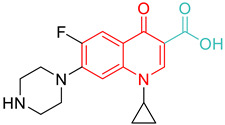		(+)/(−)	DNA Gyrase, topo IV
			Nalidixic acid		
			Cinoxacin		
	1st generation		Norfloxacin		

			Lomefloxacin		
			Enoxacin		
			Ofloxacin		
	2nd generation		Ciprofloxacin *		
			Levofloxacin		

			Sparfloxacin		
			Gatifloxacin		
	3rd generation		Moxifloxacin		

	4th generation		Trovafloxacin		
9	Streptogramins	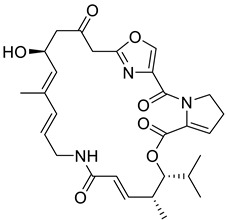	Quinupristin Daflopristid Pristinamycin IIA * Pristinamycin IA	(+)	Protein synthesis
10	Sulphonamides	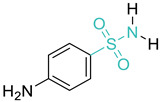	Prontosil Sulphanilamide * Sulfadiazine Sulfisoxazole Sulfamethoxazole Sulfathalidine	(+)/(−)	Folate synthesis
11	Tetracyclines	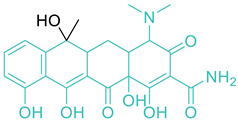	Tetracycline* Chlortetracycline Demeclocycline Minocycline Oxytetracycline Methacycline Doxycycline	(+)/(−)	Protein synthesis
12	Others	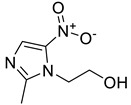	Metronidazole* Polymyxin B Trimethoprim	(+)/(−) (−) (+)/(−)	Protein synthesis Cell wall Cell Wall

* indicates the molecule shown, with the core structures/scaffolds of each category colored in light green, except for quinolones, which are colored red.

**Table 2 biomedicines-11-00371-t002:** MIC (µg/mL) values of compound **13a–f**.

Entry	Compd.	*E. coli*	*S. typhimurium*	*S. enteritidis*	*P. aeruginosa*
1	**13a**	0.5	2	1	2
2	**13b**	4	2	2	2
3	**13c**	4	8	4	16
4	**13d**	4	8	8	128
5	**13e**	2	16	8	128
6	**13f**	2	2	2	8
7	**S2**	0.125	0.125	0.0625	1

**Table 3 biomedicines-11-00371-t003:** Antibacterial and DNA gyrase inhibitory properties of compounds **28–37**.

Entry	Compounds	(MIC, µM)	DNA Gyrase (IC_50_, mg/L)	
		FQR *E. coli*	Wild-Type	GyrA Mutant
1	**18**	1.93	0.26	0.26
2	**19**	1.78	0.24	0.24
3	**20**	1.48	0.16	0.16
4	**21**	1.12	0.06	0.4
5	**22**	1.03	0.12	1.2
6	**23**	3.19	0.12	0.8
7	**24**	3.59	0.18	1.8
8	**25**	2.05	0.18	2.0
9	**26**	2.67	0.08	0.6
10	**27**	1.85	0.22	2.2
11	**S3**	22.76	0.30	>200
12	**S4**	23.70	-	-
13	**S5**	22.41	-	-

**S3**–**S5**: Standard drugs (Figure 2).

**Table 4 biomedicines-11-00371-t004:** Predicted drug-like properties of the quinolones in this study.

Entry	Compds.	MW	logP	HBD	HBA	TPSA	Rotatable Bonds	hERG pIC_50_	BBB	HIA
1	**S1**	331.3	−1.08	2	6	74.6	3	4.44	−	+
2	**S2**	319.3	−1.03	2	6	74.6	3	4.25	−	+
3	**S3**	375.4	−0.74	2	7	83.8	4	4.41	−	+
4	**S4**	361.4	0.6296	1	7	83.8	4	4.74	−	+
5	**S5**	361.4	0.55	1	7	75.0	2	4.76	−	+
6	**S6**	401.4	−0.45	2	7	83.8	4	4.67	−	+
7	**S7**	389.4	−1.12	2	9	123	5	3.92	−	+
8	**S8**	440.8	1.65	3	8	121.7	3	4.52	−	+
9	**1**	593.0	4.51	1	10	120.8	6	5.85	−	+
10	**2**	473.4	1.28	2	13	172.7	4	3.87	−	+
11	**3**	361.4	−1.11	2	7	83.8	4	4.41	−	+
12	**4**	433.4	−0.73	2	9	110.1	7	4.29	−	+
13	**5**	620.5	5.4	1	8	83.6	7	6.65	−	+
14	**6**	500.6	2.95	2	6	94.5	7	5.00	−	+
15	**7**	474.5	2.71	2	6	94.5	6	5.16	−	+
16	**8**	461.5	3.25	2	8	94.5	5	5.58	−	+
17	**9**	288.3	1.28	1	8	99.2	2	4.10	−	+
18	**10**	302.3	4.67	1	8	99.2	3	4.23	−	+
19	**11**	528.5	2.27	2	10	115.4	4	4.03	−	+
20	**12**	508.5	1.74	2	10	115.4	4	4.05	−	+
21	**13**	564.6	2.45	1	10	120.2	7	4.58	−	+
22	**14**	459.5	2.44	1	8	99.9	8	4.91	−	+
23	**15**	459.5	2.44	1	8	99.9	8	4.95	−	+
24	**16**	1081	4.38	4	20	243.2	16	5.29	−	+
25	**17**	1125	4.52	4	21	252.4	17	5.16	−	+
26	**18**	472.9	4.48	2	7	99.7	8	4.69	−	+
27	**19**	486.9	4.99	2	7	99.7	9	4.84	−	+
28	**20**	456.5	3.68	2	7	99.7	7	4.83	−	+
29	**21**	470.5	4.04	2	7	99.7	8	4.63	−	+
30	**22**	484.5	4.60	2	7	99.7	9	4.79	−	+
31	**23**	416.4	2.18	4	8	130.9	3	3.77	−	+
32	**24**	400.4	2.51	3	7	110.7	3	3.81	−	+
33	**25**	490.5	3.97	2	7	99.7	6	4.52	−	+
34	**26**	444.4	2.70	3	8	119.9	5	3.97	−	+
35	**27**	442.5	3.32	2	7	99.7	6	4.33	−	+
36	**28**	542.6	2.81	1	10	114.2	10	5.62	−	+
37	**29**	747.8	0.09	4	18	219.9	13	2.69	−	+
38	**30**	432.5	2.52	2	7	93.4	4	5.60	−	+
39	**31**	466.6	4.28	0	6	68.6	3	6.12	−	+
40	**32**	514.6	4.69	0	6	68.6	4	6.45	−	+
41	**33**	466.6	4.28	0	6	68.6	3	6.15	−	+
41	**34**	514.6	4.64	0	6	68.6	4	6.48	−	+
43	**35**	432.5	2.94	2	7	79.4	5	5.98	−	+
44	**36**	432.5	2.52	2	7	93.4	4	5.60	−	+
45	**37**	374.9	3.63	1	5	64.2	3	5.67	−	+
46	**38a**	293.7	1.51	0	6	65.6	2	5.33	−	+
47	**38b**	307.7	1.95	0	6	65.6	2	5.45	−	+
48	**38c**	321.7	2.34	0	6	65.6	4	5.64	−	+
49	**38d**	321.7	2.31	0	6	65.6	3	5.52	−	+
50	**39**	1159	1.51	16	26	438.3	41	1.39	−	+
51	**40**	369.3	3.88	1	3	49.9	4	5.18	+	+
52	**41**	457.6	3.11	1	7	76.6	6	6.61	−	+
53	**42**	463.5	1.77	1	8	92.4	7	6.71	−	−
54	**43**	294.3	2.67	3	5	82.1	3	4.55	−	+
55	**44**	335.4	2.91	3	4	56.9	5	5.23	+	+
56	**45**	400.3	3.41	3	4	56.9	5	5.36	+	+
57	**46**	355.8	3.16	3	4	59.9	5	5.25	+	+
58	**47**	399.4	2.92	2	8	110	5	4.68	−	−
59	**48**	177.2	0.54	3	4	73.3	0	4.24	−	+
60	**49**	546.5	1.25	2	13	163.5	13	4.21	−	+
61	**50**	595.6	2.51	1	13	145	8	4.71	−	−

## Data Availability

Not applicable.

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
