# Peer review of "DNA Gyrase as a Target for Quinolones"

_biomedicines, 2023, doi:10.3390/biomedicines11020371_

Round 1

Reviewer 1 Report

This work introduced the DNA gyrase and the action mechanism of DNA gyrase poison quinolone, and summarized the development of new quinolones targeting DNA gyrase from 2008 to 2022. However, important concerns are given as following aspects:
1. The 1st and 2nd paragraphs of introduction are verbose and especially Table 1 takes up a lot of space and is unnecessary. In Table 1, the general formula of the chemical core structure/scaffold of drug is a little unreasonable, for example, some structures do not draw R substituents; at the part of Entry 3
β-Lactams, penicillin is a five-membered ring structure and cephalosporin is a six-membered ring structure; (+)/(-) was not commented on the second page.
2. At the part of “Synthetic 4-quinolones as potential DNA gyrase poisons” and “Synthetic 2-quinolones as potential DNA gyrase inhibitors”, the authors simply listed the contents without obvious logical order, just in chronological order. There is a suggestion to list by site modification of quinolone. Moreover, the authors described that both 4-quinolones and 2-quinolones could act on DNA gyrase. However, due to the distinction in their structures,  the differences in the specific mechanisms of action was not clearly explained in the manuscript. On page 10, compounds 4 and 5 do not contain the quinone structure of quinolone classical C-3/C-4, so reasons for why do they had strong DNA cyclotronase inhibitory activity should be presented.
3. The 4th paragraph refers to “In general, two mechanisms are responsible for the antibacterial properties of drugs targeting DNA gyrase. One is the inhibition of the enzymatic activity of DNA gyrase (catalytic inhibitors), and the second is termed gyrase poisoning because the drugs stabilize the covalent enzyme–DNA complex which can ultimately be lethal to the cell”. However, this review wants to summarize quinolones as DNA gyrase poisons instead of DNA gyrase inhibitors, but many of the new quinolones should belong to inhibitors, such as reference 62. There is a suggestion to summarize new quinolones as both DNA gyrase poisons and DNA gyrase inhibitors.
4. The part of “Drug-like properties” is redundant and unnecessary, from which we cannot find summative drug-like properties.
5. The 2nd paragraph of conclusion for development trend of quinolone modification targeting DNA gyrase be more detailed and useful.
6. Various errors were presented in the manuscript:
1) The title of Table 1 should be a separate paragraph.
2) “2. DNA gyrase structure” at the end of the 4th paragraph of introduction should be deleted.
3) “3. Synthetic 2-quinolones as potential DNA gyrase inhibitors” should be “5. Synthetic 2-quinolones as potential DNA gyrase inhibitors”. The following titles should also be updated.
4) Some minor mistakes need to be corrected: 1. Compound in Table should be compounds. 2. In Figure 4. , “Bioavalability” should be “Bioavailability”.
5) Initial format of references was inconsistent, and the names of bacteria and some words should be italic.
6) There were a lot of tense problems throughout the whole manuscript.
7) “Figure4” should be presented as “Figure 4”, “Compd.” should be “Compds.”, and “<5” is recommended as “< 5”.
More work is needed before publication. It is recommended for publication in Biomedicines after major revision.

Author Response

Date:               January 20, 2023

Subject:           Revised Manuscript ID: biomedicines-2164680 

Dear Reviewer,

Thank you for reviewing our manuscript. We appreciate your helpful and valuable suggestions. We have reviewed them carefully and believe we have made all appropriate changes to our manuscript.

Comments and Suggestions for Authors:

This work introduced the DNA gyrase and the action mechanism of DNA gyrase poison quinolone and summarized the development of new quinolones targeting DNA gyrase from 2008 to 2022. However, important concerns are given as following aspects:

Comment 1

The 1st and 2nd paragraphs of introduction are verbose and especially Table 1 takes up a lot of space and is unnecessary. In Table 1, the general formula of the chemical core structure/scaffold of drug is a little unreasonable, for example, some structures do not draw R substituents; at the part of Entry 3 β-Lactams, penicillin is a five-membered ring structure and cephalosporin is a six-membered ring structure; (+)/(-) was not commented on the second page.

Response

We revised the introduction to make it less verbose, particularly paragraph 2. We have also corrected the entry in the table as suggested.

Comment 2

At the part of “Synthetic 4-quinolones as potential DNA gyrase poisons” and “Synthetic 2-quinolones as potential DNA gyrase inhibitors”, the authors simply listed the contents without obvious logical order, just in chronological order. There is a suggestion to list by site modification of quinolone. Moreover, the authors described that both 4-quinolones and 2-quinolones could act on DNA gyrase. However, due to the distinction in their structures, the differences in the specific mechanisms of action was not clearly explained in the manuscript. On page 10, compounds 4 and 5 do not contain the quinone structure of quinolone classical C-3/C-4, so reasons for why they had strong DNA cyclotronase inhibitory activity should be presented.

Response

We tried our best to reorganize the content as suggested. Now we have subcategorized the 4-quinolone section as

4-Quinolones with free -COOH group at C-3

4-Quinolones lacking free -COOH group at C-3

Structure 4 and 5 (now structure number 31 and 32) are fused structures containing the 4-quinolone scaffold. As the focus of the reported article is on 4-quinolone, we thought to include in our collection.

If the reviewer or editor believes they are not suitable to include, we are open to taking them out.

Comment 3

The 4th paragraph refers to “In general, two mechanisms are responsible for the antibacterial properties of drugs targeting DNA gyrase. One is the inhibition of the enzymatic activity of DNA gyrase (catalytic inhibitors), and the second is termed gyrase poisoning because the drugs stabilize the covalent enzyme–DNA complex which can ultimately be lethal to the cell”. However, this review wants to summarize quinolones as DNA gyrase poisons instead of DNA gyrase inhibitors, but many of the new quinolones should belong to inhibitors, such as reference 62. There is a suggestion to summarize new quinolones as both DNA gyrase poisons and DNA gyrase inhibitors.

Response

Thank you for the comment. We do agree with your point but most of the chemistry-based research articles do not discuss DNA gyrase poisoning and inhibitor differently. As quinolones are known to act as DNA gyrase poisons in addition to acting as catalytic inhibitors. We have included this statement with relevant references. We hope this statement will avoid confusion.

Comment 4

The part of “Drug-like properties” is redundant and unnecessary, from which we cannot find summative drug-like properties.

Response

We have now included a paragraph summarizing the findings.

Comment 5

The 2nd paragraph of the conclusion for the development trend of quinolone modification targeting DNA gyrase be more detailed and useful.

Response

We have now revised the conclusion part as suggested.

Comment 6

Various errors were presented in the manuscript:1) The title of Table 1 should be a separate paragraph.2) “2. DNA gyrase structure” at the end of the 4th paragraph of introduction should be deleted.3) “3. Synthetic 2-quinolones as potential DNA gyrase inhibitors” should be “5. Synthetic 2-quinolones as potential DNA gyrase inhibitors”. The following titles should also be updated.4) Some minor mistakes need to be corrected: 1. Compound in Table should be compounds. 2. InFigure 4. , “Bioavalability” should be “Bioavailability”.5) Initial format of references was inconsistent, and the names of bacteria and some words should be italic.6) There were a lot of tense problems throughout the whole manuscript.7) “Figure4” should be presented as “Figure 4”, “Compd.” should be “Compds.”, and “<5” is recommended as “< 5”.

Response

Thank you for all your constructive suggestions. We have carefully checked the manuscript and corrected all possible errors including the suggested ones.

Reviewer 2 Report

The review  DNA Gyrase as a Target for the Quinolones after consideration of the following major comments.

1)      DNA gyrase and DNA topoisomerase what is the differences between , should add to abstract.

2)      Table1 , has no relation with the pervious paragraph so why authors added.\

3)      Page 10, compounds 6 and 7 are not quinolone derivatives ( no C=O).

4)      Page 15, authors mentioned that methoxy and ethoxy derivatives are more potent than fluroquinolone so authors should explain?

5)      Page 20, what is the impact of 2-quinolone as antibacterial?

6)      Conclusion is a repeat of reported fundings so it must be improved.

Author Response

Date:               January 20, 2023

Subject:           Revised Manuscript ID: biomedicines-2164680 

Dear Reviewer,

Thank you for reviewing our manuscript. We appreciate your helpful and valuable suggestions. We have reviewed them carefully and believe we have made all appropriate changes to our manuscript.

Comments and Suggestions for Authors:

The review DNA Gyrase as a Target for the Quinolones after consideration of the following major comments.

Comment 1

DNA gyrase and DNA topoisomerase what is the differences between, should add to abstract.

Response

We have now included the information in the abstract.

Comment 2

Table1, has no relation with the previous paragraph so why authors added.

Response

We have now rearranged and updated the content to make it more readable.

Comment 3

Page 10, compounds 6 and 7 are not quinolone derivatives (no C=O).

Response

Compounds 6 and 7 (now numbered 33 and 34) are examples of 2-quinolones. We have colored the structures pink instead of red to follow easily.

Comment 4

Page 15, authors mentioned that methoxy and ethoxy derivatives are more potent than fluroquinolone so authors should explain?

Response

We have now explained it with a justification in the manuscript.

Further structure-activity relationship confirms the importance of methoxy and ethoxy groups for the potency. Surprisingly, 6-fluoro substituted analogs do not show an increase in biological activity but a drastic decrease.

Comment 5

Page 20, what is the impact of 2-quinolone as antibacterial?

Response

We have now included a statement on the impact of 2-quinolones as antibacterial agents.

4-Quinolone is a well-established scaffold for antibiotics and plays a crucial role in its antibacterial properties. However, many 2-quinolones which are isomeric to 4-quinolones and isosteric to coumarins were well investigated for various pharmacological properties including antibacterial. The assumption is that 2-quinolone works in a similar mode of action to 4-quinolones, but more molecular mechanism studies are needed to confirm the mechanism involved for antibacterial properties.

Comment 6

Conclusion is a repeat of reported fundings so it must be improved.

Response

We have now revised the conclusion part as suggested.

Reviewer 3 Report

The review article “ DNA Gyrase as a Target for the Quinolones” reported by A.C. Spencer and S.S. Panda reported the synthesized quinolones derivatives as antibacterial agents that target DNA Gyrase. The review article is well-designed and well-presented, which qualifies for publication in this journal. However, there are some minor comments below authors should address before acceptance.

1.      Authors are suggested to provide more detail on the biological potential of quinolones in the introduction part.

2.      Authors cited reference 17 three times in the same paragraph on page 5. Please remove the repetition.

3.      Activities of compound 18 were provided in mM units. Is it milimolar or micro molar? Please recheck it.

4.      For structures 8 and 9 authors are advised to provide activity values for comparison.

5.      Page 12, paragraph two, why do authors mention compound 11?

6.      What is the R group and its position on the ring for structure 25? Please provide its IC50 values

7.      Authors did not provide activity profile details regarding SAR for structures 30-39.

8.      Please check reference 15 numbers, is it S73-S78 or 573-578? 

Author Response

Date:               January 20, 2023

Subject:           Revised Manuscript ID: biomedicines-2164680 

Dear Reviewer,

Thank you for reviewing our manuscript. We appreciate your helpful and valuable suggestions. We have reviewed them carefully and believe we have made all appropriate changes to our manuscript.

Comments and Suggestions for Authors:

The review article “DNA Gyrase as a Target for the Quinolones” reported by A.C. Spencer and S.S. Panda reported the synthesized quinolones derivatives as antibacterial agents that target DNA Gyrase. The review article is well-designed and well-presented, which qualifies for publication in this journal. However, there are some minor comments below authors should address before acceptance.

Comment 1

Authors are suggested to provide more detail on the biological potential of quinolones in the introduction part.

Response

We have now updated the introduction part.

Comment 2

Authors cited reference 17 three times in the same paragraph on page 5. Please remove the repetition.

Response

We have now removed the repeated reference number.

Comment 3

Activities of compound 18 were provided in mM units. Is it milimolar or micromolar? Please recheck it.

Response

Thank you for checking. We have now corrected the error.

Comment 4

For structures 8 and 9 authors are advised to provide activity values for comparison.

Response

We have now included the MIC values.

Comment 5

Page 12, paragraph two, why do authors mention compound 11?

Response

We have now corrected the error.

Comment 6

What is the R group and its position on the ring for structure 25? Please provide its IC values

Response

We have now updated the structure. Provided the MIC values.

Comment 7

Authors did not provide activity profile details regarding SAR for structures 30-39.

Response

The authors used 3D-QSAR approach to identify the lead compounds and we have included the information in the manuscript.

Comment 8

Please check reference 15 numbers, is it S73-S78 or 573-578?

Response

Yes, the reference is correct.

Reviewer 4 Report

Manuscript Number: biomedicines-2164680

The manuscript entitled: DNA Gyrase as a Target for the Quinolones

This is an important scientific study, done thoroughly and expressed concisely. Therefore, the manuscript is suitable for Biomedicines in the present form.

Author Response

Date:               January 20, 2023

Subject:           Revised Manuscript ID: biomedicines-2164680 

Dear Reviewer,

Thank you for reviewing our manuscript.

Comments and Suggestions for Authors:

This is an important scientific study, done thoroughly and expressed concisely. Therefore, the manuscript is suitable for Biomedicines in the present form.

Response

We have updated our manuscript based on other reviewers’ comments and suggestions.

Round 2

Reviewer 2 Report

the article is accepted in the present form